# Can Vision Language Models Track a Heartbeat?
# A Benchmark on Frame-Level Echocardiogram Understanding

**Dingming Liu**[1] [ID]                                        DINGMING.LIU@CHARITE.DE
**Nabil Jabareen**[*1]                                         NABIL.JABAREEN@GMAIL.COM
**Soeren Lukassen**[*1]                                     SOEREN.LUKASSEN@BIH–CHARITE.DE
[1] *Center of Digital Health, Berlin Institute of Health at Charité – Universitätsmedizin Berlin, Berlin, Germany*

**Editors:** Accepted for publication at MIDL 2026

## Abstract

Echocardiogram videos are among the most common and clinically vital imaging modalities in cardiovascular medicine. They capture dynamic cardiac motion, and their accurate functional assessment requires frame-level temporal precision. Ejection fraction (EF) is an essential metric for assessing cardiac function and is computed from the left-ventricular volumes at end-diastole (EDV) and end-systole (ESV), making its estimation inherently dependent on accurate frame-wise temporal reasoning. Gernal Vision Language Models (VLMs) have recently shown strong performance in general video understanding. However, whether they can reliably reason over the fine-grained temporal dynamics required for echocardiographic interpretation remains unclear.

We benchmarked six state-of-the-art open-source VLMs, Gemma-3n, LLaVA-Interleave, LLaVA-NeXT-Video 7B/34B, and Qwen3-VL 8B/32B, on the clinically motivated task of frame-level EDV/ESV localization in apical four-chamber echocardiograms. All models performed poorly on this localization task, with errors far beyond clinically acceptable tolerances, and in some cases indistinguishably from random Monte Carlo baselines. To further test whether explicit structural guidance could compensate for limited temporal reasoning, we additionally provided left-ventricular segmentation overlays as auxiliary visual input for both tasks. However, even with segmentation cues, performance gains remained negligible in this tasks. Prompting the model to focus on masked areas only, omitting any medical context, did not lead to marked improvements.

To reduce the complexity to pure size comparison, we further evaluated a simplified two-frame binary classification task in which each model must distinguish end-diastole (ED) from end-systole (ES). Despite this simplification, performance remained low for most models on original videos, only Qwen3-VL-32B reaches an accuracy of 0.711. Providing segmentation overlays and ignoring medical background knowledge only helped Qwen3-VL in both sizes reaches accuracy over 0.9, with other models resulting in random level.

This work presents the first systematic evaluation of general-purpose VLMs on echocardiogram video analysis across progressively simplified temporal reasoning tasks. Our results reveal a fundamental limitation of current VLMs in frame-level cardiac ultrasound interpretation. This work highlights the importance of medical benchmarks for VLMs and the need for domain-specific temporal modeling in future medical VLMs. To facilitate benchmarking of VLMs on echocardiogram video analysis, we make the benchmark and all associated code publicly available here.

**Keywords:** Echocardiography, Vision Language Models, Benchmarking

---

[*] Corresponding author

## 1. Introduction

Transthoracic echocardiography is the most widely used imaging modality for the assessment of cardiac disease(Lang et al., 2015) (Gillam and Marcoff, 2024). Because it is non-invasive and does not involve ionizing radiation (Gillam and Marcoff, 2024), it is often the first-line imaging method for patients with suspected cardiovascular disease(Steeds, 2011). To assess global cardiac function, the ejection fraction (EF) is typically derived from left ventricular volumes at end-diastole (EDV) and end-systole (ESV). This process requires accurately identifying the corresponding ED and ES frames in echocardiogram videos, then estimating chamber volumes from these frames. Accurate quantification of cardiac chamber size and function is therefore a cornerstone of cardiac imaging (Lang et al., 2015). In routine clinical practice, however, these measurements still rely heavily on time-consuming manual work(Zolgharni et al., 2017).

Vision-language models (VLMs) have recently demonstrated strong performance on a range of general video understanding tasks, including action recognition, temporal event localization, and video question answering (Chambon et al., 2022) (Li et al., 2024) (Team, 2025b). These advances raise the question of whether such models can be repurposed for medical video interpretation. For echocardiography in particular, a promising use case would be automating frame-level tasks such as locating EDV and ESV frames as an intermediate step towards EF estimation and downstream decision support. Before such applications can be considered, VLMs must be systematically evaluated on whether they can reliably reason over fine-grained cardiac motion at the frame level.

In this work, we focused on the clinically motivated task of frame-level EDV and ESV localization in apical four-chamber echocardiogram videos. Our primary objective is to assess whether state-of-the-art open-source VLMs can identify the frames corresponding to the largest and smallest left ventricular cavity. To this end, we defined a frame localization task (T1) in which each input subsequence was constructed to contain exactly one EDV frame and one ESV frame, and the models were prompted to output the indices of these frames within a specified index range.

To study how different forms of visual and textual guidance affect frame localization, we evaluated T1 under three levels of varying context. In the original setting, models received unmodified echocardiogram videos. In the segmented setting, models received videos with a mask highlighting the left ventricle. In the non-medical segmented setting, the same overlays were used, but models were explicitly instructed to disregard medical context and consider only the size of the masked region. This hierarchy of input conditions allowed us to test whether explicit structural cues or the removal of medical terminology could compensate for limited temporal reasoning.

In addition to this main benchmark, we included a simpler two-frame discrimination task (T2) as an auxiliary analysis. In T2, models were asked to select the EDV or ESV frame from a pair of annotated frames extracted from the same video. This auxiliary task removed long-range temporal context and reduced the problem to direct size comparison between two candidate frames, which helped interpret failures observed in the more challenging localization setting.

Across these tasks and input conditions, we systematically assessed whether current VLMs could achieve clinically meaningful frame-level performance on echocardiogram videos.

## 2. Related works

Several echocardiography-specific foundation models trained on large-scale ultrasound data have been proposed in recent years and have demonstrated strong performance on their respective target tasks. EchoCLIP adapts CLIP-style contrastive learning to align echocardiography images or videos with textual descriptions, demonstrating strong zero-shot and transfer performance on downstream tasks such as ejection fraction (EF) prediction (Christensen et al., 2024). EchoPrime further scales this paradigm to multi-view, multi-video echocardiography exams using millions of video-report pairs, and reports strong results on comprehensive study-level interpretation tasks (Vukadinovic et al., 2025). In parallel, EchoFM explores self-supervised strategy to learn general-purpose spatiotemporal representations for echocardiography, which can later be effectively fine-tuned for tasks such as EF regression or classification(Kim et al., 2024).

Despite often being described as "vision-language" models, these approaches are fundamentally representation or prediction models rather than conversational VLMs. They do not support dialogue-style interaction, instruction following, or constrained index-based outputs, and are therefore not suitable for interactive clinical scenarios. What's more, they are not designed for frame-level video understanding, such as identifying end-diastolic or end-systolic frames by returning explicit frame indices. Instead, their primary output is a scalar prediction (e.g., EF), sometimes assisted by segmentation or view classification.

As discussed above, Specialized models exist for many well-defined problems. However, as experience with LLM chatbots has shown, ease of access often leads users to apply general-purpose tools to health-related tasks(American Psychological Association, 2023). While developers of deep learning systems for medical imaging may prefer tailored architectures, end users are likely to be tempted to repurpose generalist VLMs for tasks they were never specifically designed to solve. This makes it essential to systematically benchmark whether such a strategy can be expected to work and in which settings it reliably fails.

In contrast, our work deliberately benchmarked general-purpose VLMs under controlled prompting to evaluate their ability to reason about cardiac motion at the frame level. We nevertheless included EchoCLIP-based EF prediction as a domain-specific upper bound to contextualize the gap between task-optimized echocardiography models and general purpose VLMs on standard clinical metrics.

## 3. Methods

### 3.1. Dataset

We based our experiments on a subset containing 100 videos from the EchoNet-Dynamic dataset (Ouyang et al., 2020), which contains over 10,000 apical four-chamber echocardiography videos from individuals who underwent imaging at Stanford University Hospital. For each video, the left ventricle was traced along the endocardial border at one EDV and one ESV frame within a single cardiac cycle. This provided ground-truth frame indices for a single EDV-ESV pair per video. Our subset was chosen randomly from the validation set.

In addition to the original videos, we used frame-by-frame semantic segmentation (Ouyang et al., 2020) of the left ventricle on our chosen subset videos of EchoNet-Dynamic. The segmented version of each video contained a red mask indicating the left ventricular region in

every frame, example of original frames and frames with segmentation masks are shown in Fig 1. These segmentation overlays allowed us to study whether explicit structural cues helped VLMs focus on the relevant anatomy.

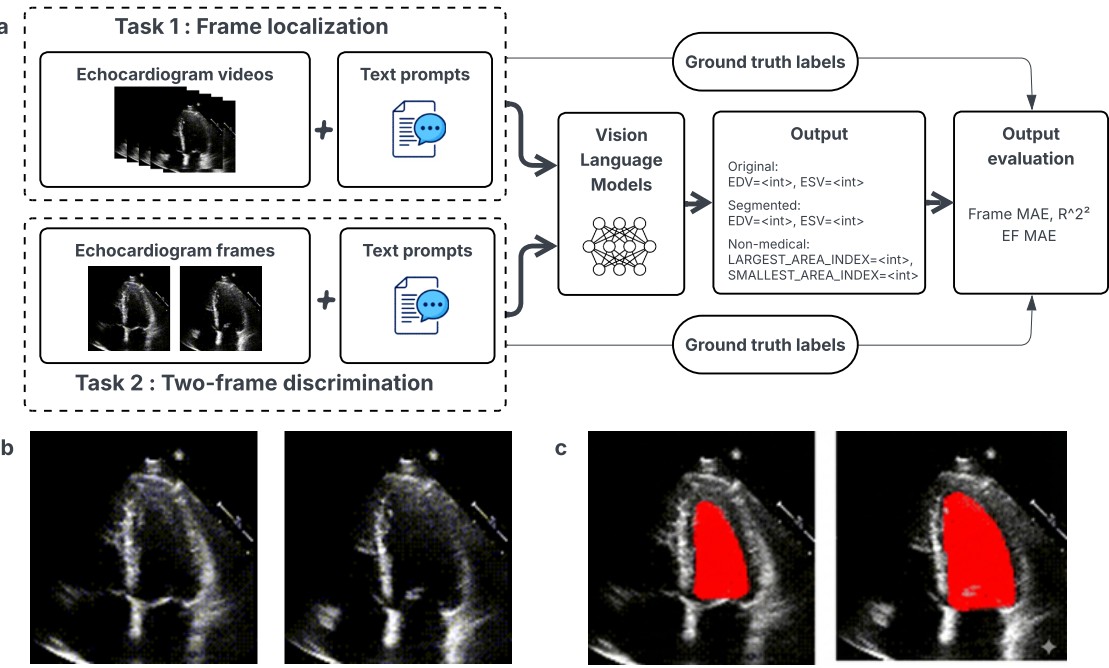

Figure 1: Task and data overview. **a**, Task overview of T1 and T2. **b**, Example of EDV and ESV frames. Left is ESV frame, with the smallest left ventricle cavity. On the right is EDV frame with the largest left ventricle cavity. **c**, Example of segmentation masks on ESV frame(left) and EDV frame(right), on which segmented and non-medical context conditions were performed.

## 3.2. Frame localization task (T1)

Our primary task is frame-level localization of EDV and ESV, as shown in Fig 1.

For each video, we extracted a contiguous subsequence corresponding to approximately one cardiac cycle, which includes one full diastolic phase followed by one full systolic phase (i.e., a complete relaxation and contraction of the heart). Let $\ell$ denote the temporal distance (in frames) between the annotated EDV and ESV frames, i.e., the EDV-to-ESV interval that approximately covers half a heartbeat. We then constructed the subsequence by taking the extracted clip spanning from $\frac{1}{3}\ell$ frames before the earlier annotated frame to $\frac{1}{3}\ell$ frames after the later annotated frame. This design guarantees that both EDV and ESV frames lie strictly inside the extracted subsequence, rather than being the first or the last frame. What's more, it also ensured that each input contained exactly one EDV-ESV pair while

reducing the temporal length of the sequence to make the task more tractable. In practice, the lengths of the subsequence range from 21 to 64 frames and are approximately distributed normally (Shapiro-Wilk test, $p = 0.17$), depending on the patient's heart rate and the video frame rate (FPS).

For each subsequence, we re-centered the frame indices to a local range $[0, \ell_i + 1)$ for subsequence length $\ell_i$. The models were informed about the number of frames and the valid index range and were required to select indices only from this range. Prompts also instructed the models to output frame indices in a predefined format. Structured prompts such as "*I sampled exactly $\ell_i$ frames from the video. Only use these $\ell_i$ frames. Find the frame indices of ESV (smallest Left Ventricle cavity) and EDV (largest Left Ventricle cavity) from input frames indexing ranging in $[0, \ell_i + 1)$. Reply exactly as: Frame: EDV = $<int>$, ESV = $<int>$*". To probe instruction sensitivity, we considered three prompting variants, asking for EDV first and ESV second, reversing this order, and requesting only a single phase (EDV or ESV) at a time.

T1 was evaluated under three different context conditions:

- **Original**. Models received the original echocardiogram frames.

- **Segmented**. Models received videos with a red mask overlay highlighting the left ventricle. Prompts explained that "the left ventricle is marked as the red region in all frames" and that EDV and ESV correspond to the frames with the largest and smallest red region area.

- **Non-medical**. Models received the same segmented videos but were explicitly instructed to ignore medical context and to focus solely on the size of the red region, for example by asking for "the frame with the largest red region".

Across the three context conditions, the subsequence boundaries (i.e., start and end frames) were kept fixed for each video to ensure a fair comparison. More importantly, we maximized prompt consistency across conditions by keeping the instruction format identical. The EDV and ESV concept explanations, which only applies to T1 original and T1 segmented, the same in these two conditions.

For the T1 segmented context, we only introduced minimal, segmentation-specific guidance by explicitly stating that *"Left Ventricle is marked as red region in all the frames."* To further emphasize the segmentation mask as the key visual cue, we additionally included: *"Decide which frame is ESV/EDV, that has the smallest/largest red region area."* For the T1 non-medical context, we similarly restricted modifications to the smallest extent necessary by instructing the model to ignore irrelevant semantic information: *"Ignore all background details, anatomy, or medical context. Focus on the red region drawn on top of the images."* Instead of directly mentioning EDV or ESV, we reformulated the target as identifying the frame *"with the LARGEST and SMALLEST red area"*, which preserves the same underlying decision rule while removing medical terminology.

Together, these settings defined the main benchmark conditions of T1 on original, segmented, and non-medical inputs. For the full prompts see here.

### 3.3. Auxiliary two-frame discrimination task (T2)

To complement the localization benchmark, we defined an auxiliary two-frame discrimination task, shown in Fig 1. For each video, we extracted the annotated EDV and ESV frames and formed a short two-frame sequence. In separate runs, each model was instructed to identify either the EDV frame or the ESV frame. The same three context conditions as in T1 were considered.

T2 removed the need for long-range temporal integration and transferred the problem to direct comparison of cavity size between two candidate frames. We used this task primarily as a diagnostic tool to assess whether failures on T1 were due to temporal reasoning, difficulty in attending to the relevant regions, or more basic limitations in comparing differences in cavity size.

### 3.4. Models and evaluation pipeline

We benchmarked six state-of-the-art open-source VLMs that support video or multi-image input, namely LLaVA-Interleave-7B(Li et al., 2024), LLaVA-NeXT-Video-7B, LLaVA-NeXT-Video-34B (Zhang et al., 2024)(Liu et al., 2024), Qwen3-VL-8B-Instruct, Qwen3-VL-32B-Instruct(Team, 2025b)(Bai et al., 2023), Gemma-3n-E4B-IT(Instruct)(Team, 2025a).

For T1 original, we additioinally performed a specialized echocardiogram model EchoCLIP (Christensen et al., 2024) on the same videos for EF prediction, providing an upper bound for expected model performance.

All models were accessed via Hugging Face. For each task and input modality, all models received semantically equivalent text instructions, adapted only as needed to match model-specific formatting requirements.

Because of architectural differences, models accepted video input in different forms. Gemma-3n and LLaVA-Interleave processed multiple images, which were provided as individual frames sampled from the same video subsequence. LLaVA-NeXT-Video and Qwen3-VL processed a single video tensor directly, which were fed with temporally ordered frames.

For each model and experimental condition, we ran ten different random seeds. For each seed, three independent passes were performed per video and the predictions were aggregated to reduce stochastic variability. For T1, we reported frame-level mean absolute error (MAE) and correlation-based $R^2$ of frame indices between predictions and ground truth. For a better understanding of the model performance, we further calculated the EF from the volumes of the predicted EDV and ESV frames by $EF = \dfrac{EDV - ESV}{EDV}$ for each video. MAE was also calculated from calculated EF and ground truth EF. For T2, we reported classification accuracy.

### 3.5. Monte Carlo random baselines

To quantify improvements over chance, we estimated random baselines using Monte Carlo simulation for both localization and binary discrimination tasks.

For T1, each video and each condition specified a searchable frame index range $[0, i_{\max})$ and a ground-truth EDV or ESV frame index $i_{gt}$. In each Monte Carlo trial, we sampled a single integer frame index uniformly at random from this range for each video and computed the MAE of the index error averaged over all videos. Repeating this procedure $T = 10{,}000$

times yielded an empirical distribution of random MAE values from which we derived the mean random MAE and its 95% confidence interval.

For T2, each evaluation run consisted of 100 two-frame samples with a binary ground-truth label vector $y \in \{0, 1\}^{100}$ that indicated which frame was EDV or ESV. Repeating this process $T = 10,000$ times per run yielded an empirical distribution of random accuracies. We reported the mean and 95% confidence intervals of these random baselines alongside model accuracies.

## 4. Experiments and Results

In this section, we present a comprehensive evaluation of six open-source VLMs on the frame localization task across different context conditions, followed by a brief analysis of the auxiliary two-frame discrimination task. Unless otherwise specified, we focus on frame-level mean absolute error (MAE) for T1 and classification accuracy for T2.

Across all T1 settings, we found that current VLMs struggled substantially, with localization errors far beyond clinically acceptable tolerances and often close to random performance.

| Model | Output format | EDV (frame) | | ESV (frame) | | EF (%) |
| | | MAE↓ | $R^2 \uparrow$ | MAE↓ | $R^2 \uparrow$ | MAE↓ |
|---|---|---|---|---|---|---|
| Random | - | 11.87 | - | 11.62 | - | - |
| LLaVA-Interleave | edv_esv | $100.68 \pm 93.86$ | $0.02 \pm 0.04$ | $85.28 \pm 118.20$ | $0.08 \pm 0.13$ | $51.49 \pm 2.62$ |
| | esv_edv | $46.04 \pm 69.92$ | $0.05 \pm 0.06$ | $38.63 \pm 66.95$ | $0.04 \pm 0.05$ | $65.32 \pm 3.09$ |
| | edv | $7{,}538.39 \pm 23{,}615.93$ | $0.03 \pm 0.03$ | - | - | - |
| | esv | - | - | $15.20 \pm 1.66$ | $0.08 \pm 0.09$ | - |
| LLaVA-NeXT-Video-7B | edv_esv | $25.72 \pm 2.67$ | $0.01 \pm 0.01$ | $20.75 \pm 3.39$ | $0.03 \pm 0.04$ | $34.36 \pm 2.41$ |
| | esv_edv | $28.14 \pm 4.51$ | $0.03 \pm 0.03$ | $22.58 \pm 5.81$ | $0.03 \pm 0.03$ | $52.66 \pm 3.61$ |
| | edv | $33.14 \pm 6.37$ | $0.04 \pm 0.03$ | - | - | - |
| | esv | - | - | $28.30 \pm 6.03$ | $0.03 \pm 0.03$ | - |
| LLaVA-NeXT-Video-34B | edv_esv | $10.09 \pm 0.86$ | $0.17 \pm 0.07$ | $\mathbf{6.29 \pm 0.47}$ | $0.37 \pm 0.09$ | $37.04 \pm 2.72$ |
| | esv_edv | $16.49 \pm 0.67$ | $0.33 \pm 0.14$ | $10.76 \pm 0.69$ | $0.15 \pm 0.08$ | $60.29 \pm 1.09$ |
| | edv | $8.52 \pm 0.59$ | $0.05 \pm 0.05$ | - | - | - |
| | esv | - | - | $10.96 \pm 0.57$ | $0.17 \pm 0.07$ | - |
| Gemma-3n | edv_esv | $\mathbf{7.65 \pm 0.20}$ | $\mathbf{0.29 \pm 0.04}$ | $10.31 \pm 0.28$ | $0.29 \pm 0.05$ | $41.31 \pm 1.11$ |
| | esv_edv | $18.85 \pm 0.34$ | $\mathbf{0.59 \pm 0.06}$ | $11.29 \pm 0.40$ | $0.14 \pm 0.04$ | $75.02 \pm 3.36$ |
| | edv | $9.68 \pm 0.20$ | $\mathbf{0.53 \pm 0.03}$ | - | - | - |
| | esv | - | - | $\mathbf{4.34 \pm 0.18}$ | $\mathbf{0.57 \pm 0.06}$ | - |
| Qwen3-VL-8B | edv_esv | $8.02 \pm 0.26$ | $0.19 \pm 0.08$ | $7.06 \pm 0.36$ | $\mathbf{0.76 \pm 0.07}$ | $28.15 \pm 1.58$ |
| | esv_edv | $\mathbf{7.12 \pm 0.27}$ | $0.01 \pm 0.01$ | $\underline{\mathbf{10.32 \pm 0.20}}$ | $\underline{\mathbf{0.24 \pm 0.02}}$ | $\underline{\mathbf{36.57 \pm 6.53}}$ |
| | edv | $\underline{\mathbf{6.55 \pm 0.35}}$ | $0.06 \pm 0.04$ | - | - | - |
| | esv | - | - | $11.34 \pm 0.23$ | $0.09 \pm 0.06$ | - |
| Qwen3-VL-32B | edv_esv | $10.05 \pm 0.19$ | $0.03 \pm 0.03$ | $7.83 \pm 0.15$ | $0.54 \pm 0.12$ | $\underline{\mathbf{25.73 \pm 0.61}}$ |
| | esv_edv | $12.13 \pm 0.53$ | $0.05 \pm 0.03$ | $18.21 \pm 0.64$ | $0.01 \pm 0.01$ | $59.16 \pm 6.87$ |
| | edv | $6.64 \pm 0.30$ | $0.10 \pm 0.05$ | - | - | - |
| | esv | - | - | $11.22 \pm 0.64$ | $0.25 \pm 0.15$ | - |
| EchoCLIP | - | - | - | - | - | 8.54 |

Table 1: Metrics of model performance in T1 original. The standard deviations shown here were calculated across random seeds, showing model stochasticity. We highlighted the best model performance in different context conditions.

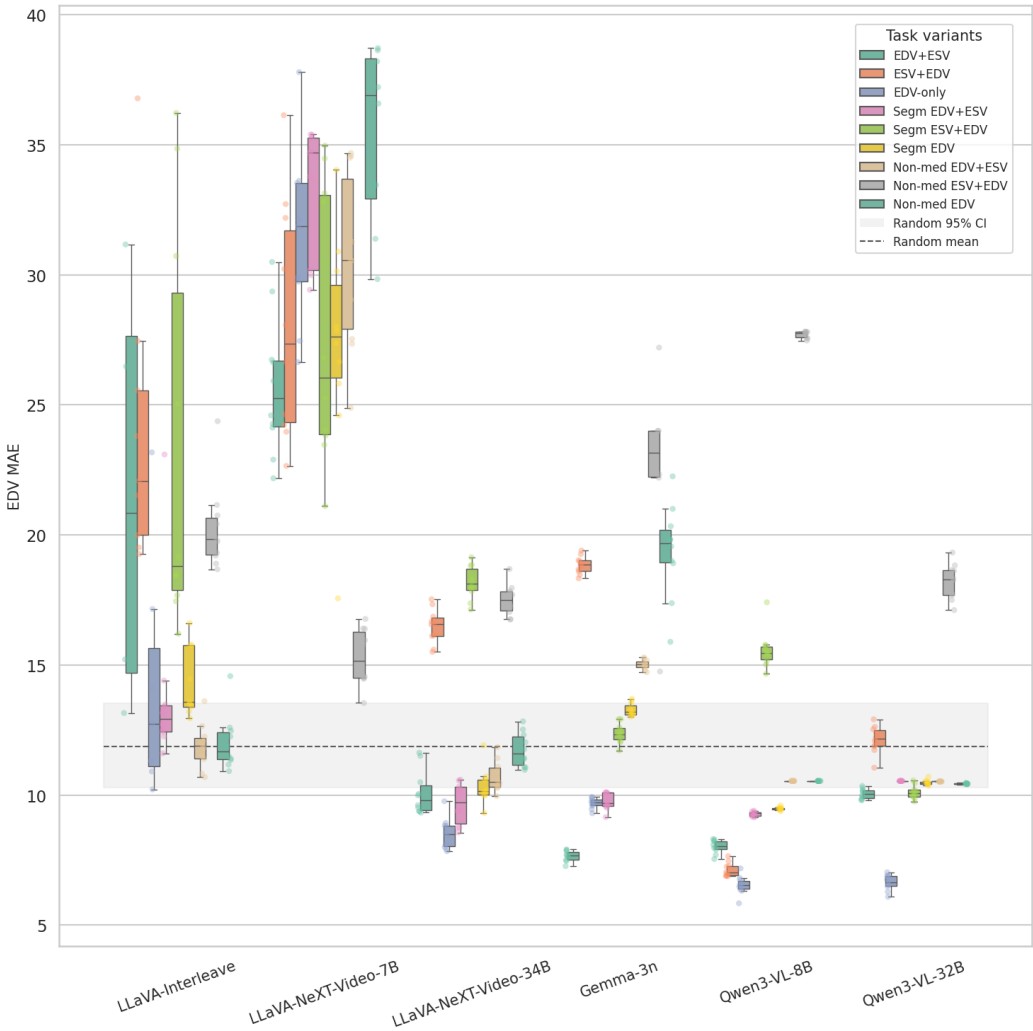

Figure 2: EDV frame localization (T1) performance in MAE across different task settings. The dashed line indicates the Monte Carlo random baseline. Qwen3-VL-8B achieved the best performance in four out of all nine task variants.

## 4.1. T1 on different settings

We first evaluated all models on T1 using original echocardiogram videos as defined in Section 3.2. The results are summarized in Table 1, with standard deviations calculated across different trials that show the stochasticity of the model (random-seed variance). Table 5 (see in Appendix A) shows the variance among samples. The "Output format" column in the tables encodes the prompting variants described in Section 3.2. For T1 original and T1 segmented, **edv_esv** indicates prompts that asked for EDV first and ESV second, **esv_edv** denotes the reversed order, and **edv/esv** refer to prompts that requested only one single

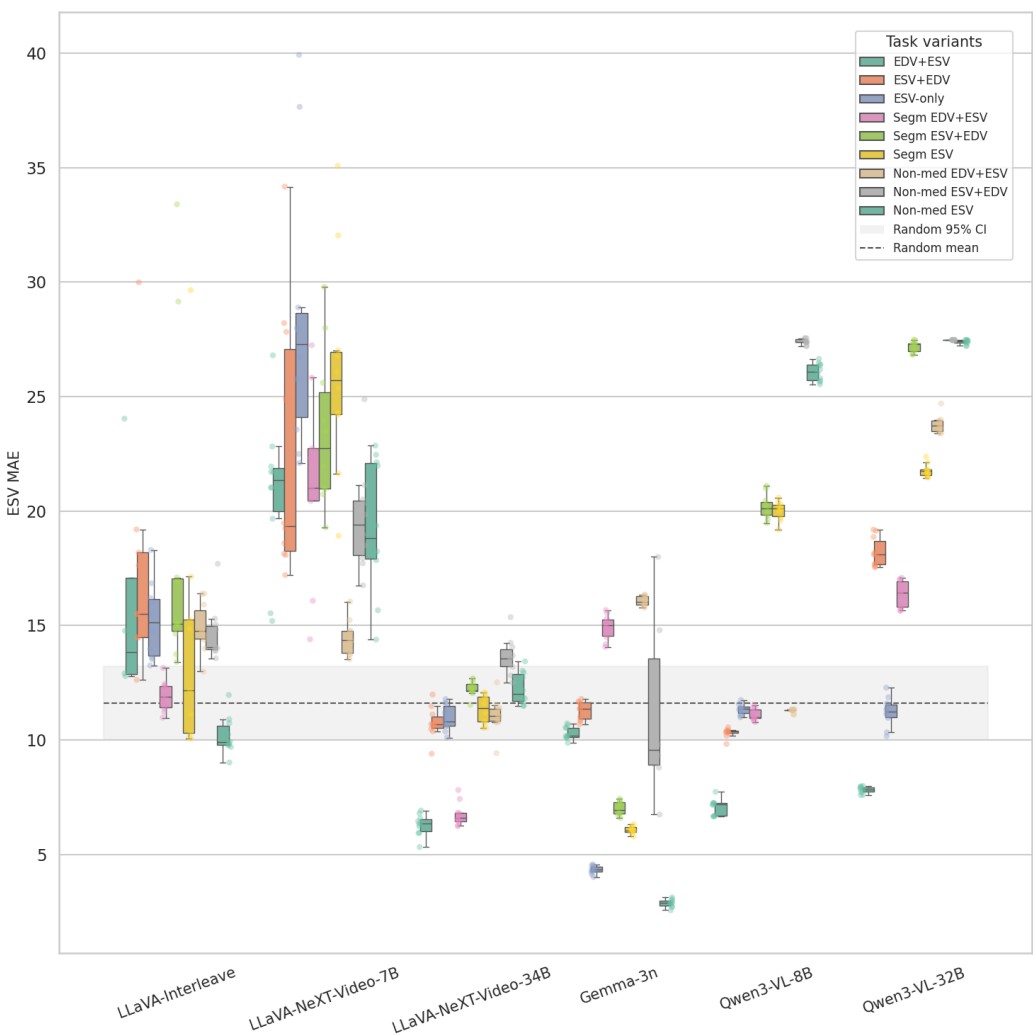

Figure 3: ESV frame localization (T1) performance in MAE across different task settings. The dashed line indicates the Monte Carlo random baseline. Gemma-3n achieved the best performance in five out of all nine task variants.

phase. For T1 non-medical, which does not involve any explicit medical terminology, we followed the same design principle and used `large_small` for prompts that first asked for the frame with the largest red area and then the smallest, `small_large` for the reversed order, and `large`/`small` for single-phase prompts that requested only one of the two.

In the setting where EDV was requested first and ESV second, which reflects the typical order in clinical practice, Gemma-3n achieved the best in EDV localization and LLaVA-NeXT-Video-34B had the best score in ESV localization. Qwen3-VL-8B outperformed in the ESV first and EDV second setting with the best scores in both EDV and ESV predictions. These models had also the best performances in ESV-only (Gemma-3n: 4.34

frames) and EDV-only (Qwen3-VL-8B: 6.55 frames) tasks (Figure 2, Figure 3). However, these errors were still larger compared to typical human variability, which is often within 2 to 4 frames (Zolgharni et al., 2017). The calculated MAE of EF further proved this, even the model with the best EF prediction failed. EchoCLIP was also performed on these original videos for EF prediction, with MAE of 8.54% shown in Table 1, far outperforming the best-performing general VLM. The remaining models exhibited even larger errors and, in some conditions, performed only marginally better than the Monte Carlo random baseline (Figure 2, Figure 3).

Overall, we did not observe a consistent pattern of model performance across prompting variants. Instead, each model's performance remained unpredictable across settings, supporting the suspicion that the models did not truly understand the videos but were effectively outputting frame indices close to chance.

| | | EDV (frame) | | ESV (frame) | | EF (%) |
|---|---|---|---|---|---|---|
| **Model** | **Output format** | **MAE↓** | $R^2 \uparrow$ | **MAE↓** | $R^2 \uparrow$ | **MAE↓** |
| Random | - | 11.87 | - | 11.62 | - | - |
| LLaVA-Interleave | edv_esv | $13.90 \pm 3.32$ | $0.15 \pm 0.08$ | $11.93 \pm 0.66$ | $0.22 \pm 0.10$ | $49.42 \pm 2.60$ |
| | esv_edv | $23.42 \pm 7.74$ | $0.20 \pm 0.17$ | $18.36 \pm 6.98$ | $0.08 \pm 0.06$ | $62.33 \pm 1.96$ |
| | edv | $18.05 \pm 11.75$ | $0.11 \pm 0.08$ | - | - | - |
| | esv | - | - | $22.02 \pm 24.40$ | $0.11 \pm 0.14$ | - |
| LLaVA-NeXT-Video-7B | edv_esv | $33.94 \pm 3.36$ | $0.02 \pm 0.02$ | $21.01 \pm 3.93$ | $0.01 \pm 0.01$ | $44.49 \pm 2.64$ |
| | esv_edv | $27.99 \pm 5.27$ | $0.02 \pm 0.02$ | $23.48 \pm 3.41$ | $0.02 \pm 0.02$ | $46.53 \pm 4.26$ |
| | edv | $27.29 \pm 4.37$ | $0.01 \pm 0.01$ | - | - | - |
| | esv | - | - | $26.14 \pm 4.66$ | $0.01 \pm 0.02$ | - |
| LLaVA-NeXT-Video-34B | edv_esv | $9.66 \pm 0.79$ | $0.07 \pm 0.04$ | $\textbf{6.75} \pm \textbf{0.51}$ | $0.44 \pm 0.10$ | $38.86 \pm 2.89$ |
| | esv_edv | $18.17 \pm 0.64$ | $\underline{\textbf{0.44} \pm \textbf{0.06}}$ | $12.24 \pm 0.34$ | $0.08 \pm 0.03$ | $67.42 \pm 2.51$ |
| | edv | $10.35 \pm 0.68$ | $0.14 \pm 0.06$ | - | - | - |
| | esv | - | - | $11.33 \pm 0.63$ | $0.12 \pm 0.07$ | - |
| Gemma-3n | edv_esv | $9.76 \pm 0.33$ | $\underline{\textbf{0.39} \pm \textbf{0.04}}$ | $14.92 \pm 0.54$ | $0.04 \pm 0.03$ | $58.56 \pm 2.85$ |
| | esv_edv | $12.37 \pm 0.38$ | $\underline{0.27 \pm 0.04}$ | $\textbf{7.02} \pm \textbf{0.31}$ | $0.10 \pm 0.03$ | $\textbf{35.39} \pm \textbf{3.25}$ |
| | edv | $13.28 \pm 0.24$ | $\underline{\textbf{0.43} \pm \textbf{0.04}}$ | - | - | - |
| | esv | - | - | $\textbf{6.08} \pm \textbf{0.15}$ | $\textbf{0.65} \pm \textbf{0.12}$ | - |
| Qwen3-VL-8B | edv_esv | $\textbf{9.27} \pm \textbf{0.09}$ | $0.05 \pm 0.02$ | $11.10 \pm 0.25$ | $\textbf{0.56} \pm \textbf{0.04}$ | $\textbf{31.63} \pm \textbf{0.59}$ |
| | esv_edv | $15.57 \pm 0.73$ | $0.01 \pm 0.01$ | $20.19 \pm 0.54$ | $\underline{\textbf{0.16} \pm \textbf{0.04}}$ | $48.88 \pm 2.46$ |
| | edv | $\underline{\textbf{9.48} \pm \textbf{0.06}}$ | $0.00 \pm 0.00$ | - | - | - |
| | esv | - | - | $20.02 \pm 0.41$ | $0.04 \pm 0.02$ | - |
| Qwen3-VL-32B | edv_esv | $10.54 \pm 0.00$ | - | $16.36 \pm 0.59$ | $0.07 \pm 0.03$ | $31.81 \pm 0.63$ |
| | esv_edv | $\underline{\textbf{10.11} \pm \textbf{0.25}}$ | $0.09 \pm 0.02$ | $27.19 \pm 0.24$ | $0.01 \pm 0.01$ | $55.90 \pm 6.84$ |
| | edv | $\underline{10.48 \pm 0.09}$ | $0.01 \pm 0.01$ | - | - | - |
| | esv | - | - | $21.78 \pm 0.29$ | $0.00 \pm 0.00$ | - |

Table 2: Metrics of model performance in T1 segmented. All of the standard deviations shown here were calculated across random seeds, showing model stochasticity. The best model performance among each output format was highlighted.

We next repeated T1 on segmented videos where the left ventricle was highlighted by a red mask, as described in Section 3.2. Full results shown in Table 2, with standard deviation calculated across different trials showing model running stochasticity(random-seed variance). Table 6 (see in Appendix A) shows the variance among samples. Contrary to our expectation, providing segmentation information did not improve performance for most models (Figure 2, Figure 3). In many cases, localization errors became even larger than on original videos.

LLaVA-NeXT-Video-34B achieved the best performance among segmented-input models when prompted to report EDV first and ESV second, while Gemma-3n performed best on the ESV-first variant and ESV only settings. However, only the ESV predictions of these settings clearly outperformed the random baseline. For the single-phase localization variants, none of the models consistently outperformed the Monte Carlo baseline, suggesting that apparent successes on joint EDV and ESV prediction may have been driven by chance.

| Model | Output format | EDV (frame) | | ESV (frame) | | EF (%) |
| | | MAE↓ | $R^2 \uparrow$ | MAE↓ | $R^2 \uparrow$ | MAE↓ |
|---|---|---|---|---|---|---|
| Random | - | 11.87 | - | 11.62 | - | - |
| LLaVA-Interleave | large_small | $11.87 \pm 0.86$ | $0.21 \pm 0.05$ | $14.91 \pm 1.07$ | $0.12 \pm 0.04$ | $53.30 \pm 3.26$ |
| | small_large | $20.25 \pm 1.66$ | $0.30 \pm 0.15$ | $14.64 \pm 1.22$ | $0.18 \pm 0.08$ | $59.58 \pm 2.22$ |
| | large | $12.01 \pm 1.06$ | $0.22 \pm 0.05$ | - | - | $45.55 \pm 1.97$ |
| | small | - | - | $10.20 \pm 0.82$ | $0.28 \pm 0.07$ | - |
| LLaVA-NeXT-Video-7B | large_small | $30.49 \pm 3.39$ | $0.03 \pm 0.03$ | $14.44 \pm 0.79$ | $0.05 \pm 0.05$ | $53.60 \pm 3.77$ |
| | small_large | $\underline{15.29 \pm 1.09}$ | $0.04 \pm 0.02$ | $19.66 \pm 2.30$ | $0.01 \pm 0.01$ | $59.49 \pm 5.10$ |
| | large | $36.69 \pm 3.97$ | $0.06 \pm 0.02$ | - | - | $47.54 \pm 4.69$ |
| | small | - | - | $19.31 \pm 2.98$ | $0.05 \pm 0.03$ | - |
| LLaVA-NeXT-Video-34B | large_small | $10.72 \pm 0.58$ | $0.16 \pm 0.05$ | $\mathbf{11.08 \pm 0.77}$ | $0.15 \pm 0.08$ | $52.07 \pm 4.47$ |
| | small_large | $17.50 \pm 0.61$ | $0.30 \pm 0.06$ | $13.64 \pm 0.80$ | $0.04 \pm 0.03$ | $66.70 \pm 5.85$ |
| | large | $11.74 \pm 0.66$ | $0.13 \pm 0.08$ | - | - | $56.11 \pm 3.58$ |
| | small | - | - | $12.27 \pm 0.71$ | $0.06 \pm 0.04$ | - |
| Gemma-3n | large_small | $15.02 \pm 0.19$ | $\mathbf{0.44 \pm 0.04}$ | $16.07 \pm 0.21$ | $0.38 \pm 0.06$ | $91.34 \pm 1.33$ |
| | small_large | $22.41 \pm 4.17$ | $0.52 \pm 0.37$ | $\mathbf{11.25 \pm 4.25}$ | $\mathbf{0.44 \pm 0.48}$ | $62.77 \pm 12.63$ |
| | large | $19.40 \pm 1.79$ | $\mathbf{0.45 \pm 0.22}$ | - | - | $40.50 \pm 7.53$ |
| | small | - | - | $\mathbf{2.88 \pm 0.17}$ | $\mathbf{0.67 \pm 0.07}$ | - |
| Qwen3-VL-8B | large_small | $10.54 \pm 0.00$ | - | $11.28 \pm 0.09$ | $\mathbf{0.66 \pm 0.08}$ | $\mathbf{31.39 \pm 0.04}$ |
| | small_large | $27.71 \pm 0.13$ | $\mathbf{0.59 \pm 0.04}$ | $27.40 \pm 0.13$ | $0.00 \pm 0.00$ | $55.71 \pm 6.77$ |
| | large | $10.54 \pm 0.00$ | - | - | - | $36.92 \pm 0.42$ |
| | small | - | - | $26.08 \pm 0.39$ | $0.01 \pm 0.02$ | - |
| Qwen3-VL-32B | large_small | $\mathbf{10.52 \pm 0.00}$ | $0.00 \pm 0.00$ | $23.79 \pm 0.39$ | $0.02 \pm 0.02$ | $37.00 \pm 0.47$ |
| | small_large | $18.20 \pm 0.68$ | $0.07 \pm 0.02$ | $27.47 \pm 0.01$ | $0.00 \pm 0.00$ | $\mathbf{54.36 \pm 0.81}$ |
| | large | $\mathbf{10.44 \pm 0.02}$ | $0.05 \pm 0.02$ | - | - | $37.60 \pm 0.12$ |
| | small | - | - | $27.39 \pm 0.10$ | $0.01 \pm 0.02$ | - |

Table 3: Metrics of model performance in T1 non-medical. The standard deviations shown here were calculated across random seeds, showing model stochasticity. The best model performance among each output format was highlighted.

To test whether medical terminology and anatomical priors distracted VLMs from the visual evidence, we performed an additional variant of T1 on segmented videos with non-medical instructions, again following the setup in Section 3.2. As results listing in Table 3, most of the models surpassed the random baselines. Similarly, Table 3 shows model stochasticity and Table 7 (see in Appendix A) shows the evaluation uncertainty. Gemma-3n, however, achieved the best MAE of 2.88 in ESV-only prediction setting, but a lot worse in EDV-only and other tasks. Overall, removing medical context and relying purely on geometric size cues did not help current VLMs perform meaningful frame localization, even when the relevant region was explicitly highlighted.

For some seeds and prompting variants, we observed degenerate behavior. The model repeatedly output the same frame index for many videos, which led to artificially low variability in predictions without reflecting meaningful localization ability. When we reversed the order in which the model was instructed to report ESV and EDV, we observed a con-

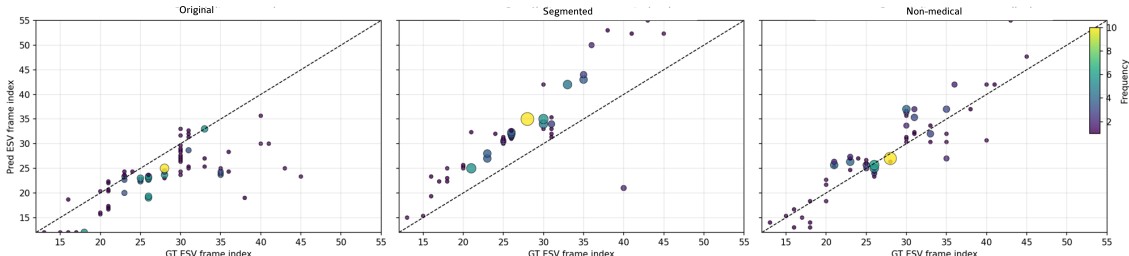

Figure 4: Prediction frame index vs ground truth frame index of Gemma-3n in ESV only output format. From left to right are T1 original, T1 segmented and T1 non-medical. There is a tendency that model tends to reply with mid-ranged indices.

sistent bias, with models tending to assign earlier frame indices to whichever term was requested first and later indices to the term requested second. Asking for only a single phase at a time reduced this order bias and led to predictions that were more evenly distributed along the diagonal when plotting ground-truth versus predicted indices. However, the overall MAEs remained high and far from clinical requirements.

We also observed systematic differences in how strictly models followed the specified index range. Smaller models often produced indices outside the admissible range, and in some cases even returned clearly nonsensical values, including negative indices or variants such as "minus zero". In contrast, the larger models, in particular LLaVA-NeXT-Video-34B and Qwen3-VL-32B, were more likely to respect the output constraints and restrict their predictions to the instructed index range.

Among all evaluated models, Gemma-3n achieved a surprisingly low MAE on the ESV-only task across different context conditions (Figure 3), which at first glance might suggest that it understood the ESV-only prediction task better than other models. However, this performance was largely driven by a simple heuristic behavior, namely a strong tendency to output frame indices near the middle of the provided index range Fig 4 (Shown only one example of Gemma-3n output, full results seen in Appendix B Figure 6, Figure 7). In clinical practice, and also in our setup, the ESV frame is typically located after the EDV frame in the cardiac cycle in order to calculate EF more precisely. This means that the ESV frame index usually lies in the later part of the sequence, so a mid-range guess can accidentally align with the true ESV position more often than pure chance would suggest. As a result, the apparently favorable MAE for Gemma-3n in the ESV-only setting does not reflect genuine frame-level understanding, but rather an artifact of this mid-range bias.

## 4.2. Auxiliary results on T2

Finally, we summarize the results of the auxiliary two-frame discrimination task. In T2, each model was asked to select the EDV or ESV frame from a pair of annotated frames extracted from the same video. This removed long-range temporal context and reduced the

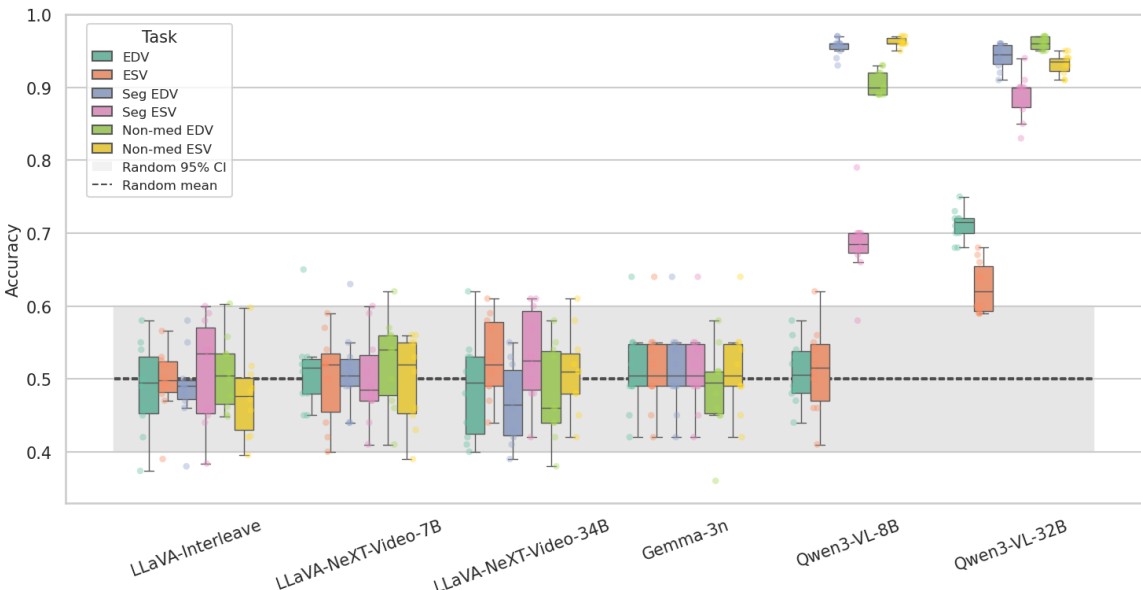

Figure 5: Binary classification (T2) accuracy for distinguishing EDV from ESV frames across different input conditions. Most models failed to substantially perform better then chance, with only Qwen3-VL-32B consistently achieving accuracies above 0.58. Segmentation overlays improved performance, particularly for Qwen3-VL models.

problem to comparing cavity size between two candidate frames. All results shown in Table 4.

On original frames (Figure 5), several models such as LLaVA-NeXT-Video in both sizes and Gemma-3n failed to achieve accuracies substantially above the random baseline of 0.5 and sometimes tended to output the same index across many videos. Qwen3-VL-32B was the only model that consistently distinguished EDV and ESV on original frames with accuracies above 0.58. Segmentation overlays improved the performance of Qwen3-VL-8B and Qwen3-VL-32B, and in the non-medical setting Qwen3-VL-32B reached accuracies above 0.9 on some sub-tasks.

These auxiliary results indicated that even when the task was simplified to a two-frame comparison, most general-purpose VLMs did not reliably distinguish EDV from ESV. Stronger models benefited from segmentation cues in this simplified setting, but this did not translate into robust frame localization performance in T1.

## 5. Discussion

Our benchmark focused on the clinically motivated task of frame-level EDV and ESV localization in apical four-chamber echocardiogram videos across multiple context conditions. The core result is that current open-source VLMs were unable to achieve clinically meaning-

| | | Accuracy | |
|---|---|---|---|
| | | EDV | ESV |
| LLaVA-Interleave | original video | $0.486 \pm 0.063$ | $0.497 \pm 0.047$ |
| | segmented | $0.489 \pm 0.053$ | $0.511 \pm 0.073$ |
| | non_medical | $0.507 \pm 0.050$ | $0.476 \pm 0.058$ |
| LLaVA-NeXT-Video-7B | original video | $0.512 \pm 0.057$ | $0.502 \pm 0.063$ |
| | segmented | $0.510 \pm 0.055$ | $0.500 \pm 0.061$ |
| | non_medical | $0.523 \pm 0.062$ | $0.499 \pm 0.062$ |
| LLaVA-NeXT-Video-34B | original video | $0.488 \pm 0.071$ | $0.528 \pm 0.057$ |
| | segmented | $0.467 \pm 0.055$ | $0.532 \pm 0.064$ |
| | non_medical | $0.480 \pm 0.065$ | $0.510 \pm 0.057$ |
| Gemma-3n | original video | $0.514 \pm 0.061$ | $0.514 \pm 0.061$ |
| | segmented | $0.514 \pm 0.061$ | $0.514 \pm 0.061$ |
| | non_medical | $0.486 \pm 0.061$ | $0.514 \pm 0.061$ |
| Qwen3-VL-8B | original video | $0.510 \pm 0.044$ | $0.513 \pm 0.060$ |
| | segmented | **$0.956 \pm 0.013$** | $0.685 \pm 0.051$ |
| | non_medical | $0.906 \pm 0.017$ | **$0.962 \pm 0.006$** |
| Qwen3-VL-32B | original video | **$0.711 \pm 0.022$** | **$0.626 \pm 0.036$** |
| | segmented | $0.942 \pm 0.018$ | **$0.888 \pm 0.032$** |
| | non_medical | **$0.961 \pm 0.009$** | $0.932 \pm 0.015$ |

Table 4: Accuracy of model performance in T2. The best accuracy in three context conditions are highlighted.

ful accuracy on this task, regardless of whether they were given original videos, segmentation overlays, or non-medical instructions.

Echocardiogram interpretation could become substantially more time-efficient if VLMs were able to assist with frame-level tasks such as EDV and ESV localization. For an experienced clinician, accurately identifying ED and ES and quantifying volumes typically takes up to a few minutes per study, with a frame-level variability on the order of 2 to 4 frames (Zolgharni et al., 2017). In contrast, all evaluated models exhibited localization errors exceeding 6 frames on average in our single-phase localization tasks. When we translated these frame errors into EF estimation, the best-case mean absolute error remained high and far from clinically acceptable thresholds. Prior work has shown that an error of just two to three frames in detecting ES can already elicit an approximate 10 percent difference in segmental ES strain (Mada et al., 2015), which underlines how demanding frame-level accuracy is for downstream functional assessment.

Our experiments further showed that naive structural cues did not solve the problem. Adding segmentation masks that highlighted the left ventricle did not systematically improve performance on frame localization and often worsened it. Removing medical context and asking models to focus solely on the red region likewise failed to yield robust improve-

ments in T1. These findings suggest that current VLMs did not automatically exploit segmentation overlays as structured guidance at the level of precision required for cardiac ultrasound.

The auxiliary two-frame discrimination task provided additional insight into these limitations. Even when temporal context was removed, but to a direct comparison between two candidate frames, many models still performed close to random. Only the strongest models benefited consistently from segmentation cues in this simplified setting, achieving high accuracies when asked to focus on the masked region. This pattern indicated that failures on T1 were not solely due to long-range temporal reasoning but also reflected a difficulty in reliably interpreting pixel-level differences in medical video frames.

Taken together, these results point to a fundamental gap between the capabilities of general VLMs and the requirements of frame-level echocardiogram understanding. General VLMs excel at semantic reasoning and coarse video understanding, but they are not trained to perform precise temporal localization of clinically relevant phases based on subtle changes in grayscale intensity. Our implementation of EchoCLIP on the T1 original setting is consistent with this picture. It suggests that pretraining on a much larger echocardiographic dataset (e.g., datasets used for training well-performing specialized models) can substantially improve downstream performance, even though it still does not fully close the gap to human experts.

Regarding clinical applicability, future developments should explicitly consider interaction with clinicians. As discussed in Section 2, existing echocardiogram-specialized models are primarily designed as representation or prediction modules rather than question-and-answer systems, which limits their usability at the point of care. A natural next step is to build on these specialized models with dialogue-like interfaces that allow clinicians to query videos in natural language and inspect intermediate outputs. At the same time, medical VLMs should be trained and calibrated to respond more conservatively, including the ability to explicitly state "I do not know" or defer to human experts when uncertainty is high, especially for critical medical decisions.

This work has several limitations. We focused on a single public dataset and a single view, so performance may differ on other acquisitions, institutions, or pathologies. We also evaluated only a fixed set of prompts and instruction formats and restricted our analysis to open-source models due to data usage constraints. Future work could explore prompt optimization, instruction tuning, and architectures tailored to periodic motion, as well as evaluations on curated datasets that can be shared with both open and closed source models.

Despite these limitations, our benchmark provides a concrete and clinically relevant stress test for VLMs in medical video analysis. It highlights that strong performance on general video understanding does not automatically translate into clinically meaningful frame-level accuracy in echocardiography. Our non-medical experiments further suggest that these failures are not only due to temporal reasoning or the specifics of medical imagery, but also reflect a broader limitation in how current VLMs handle fine-grained spatial information. We hope this work will motivate the development of medical VLMs with explicit temporal reasoning capabilities and encourage the community to design more targeted benchmarks for high-precision tasks in cardiology and beyond.

## Acknowledgments

The authors would like to thank the Charité Scientific Computing group for infrastructure support. This work was supported by the German Ministry for Research, Technology and Space (BMFTR, junior research group "Medical Omics", 01ZZ2001).

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

## Appendix A. T1 metrics with standard deviations showing Training stochasticity (random-seed variance)

| Model | Output format | EDV (frame) MAE↓ | ESV (frame) MAE↓ | EF (%) EF MAE↓ |
|---|---|---|---|---|
| Random | - | 11.87 | 11.62 | - |
| LLaVA-Interleave | edv_esv | 100.22 ± 368.63 | 84.22 ± 396.22 | 51.47 ± 20.62 |
| | esv_edv | 46.34 ± 224.91 | 38.45 ± 215.09 | 65.38 ± 26.49 |
| | edv | 7.46e3 ± 7.37e4 | - | - |
| | esv | - | 15.05 ± 5.14 | - |
| LLaVA-NeXT-Video-7B | edv_esv | 25.86 ± 16.34 | 20.90 ± 14.00 | 34.65 ± 12.06 |
| | esv_edv | 28.44 ± 18.10 | 22.57 ± 12.65 | 55.63 ± 42.60 |
| | edv | 33.30 ± 21.16 | - | - |
| | esv | - | 28.16 ± 19.13 | - |
| LLaVA-NeXT-Video-34B | edv_esv | 10.36 ± 2.87 | **6.33 ± 2.62** | 36.98 ± 14.98 |
| | esv_edv | 16.92 ± 4.80 | 10.51 ± 4.28 | 60.38 ± 22.15 |
| | edv | 8.82 ± 3.36 | - | - |
| | esv | - | 10.70 ± 4.00 | - |
| Gemma-3n | edv_esv | **7.82 ± 3.14** | 10.14 ± 4.83 | 41.31 ± 18.28 |
| | esv_edv | 19.12 ± 3.94 | 11.05 ± 6.51 | 75.02 ± 75.00 |
| | edv | 9.88 ± 2.86 | - | - |
| | esv | - | **4.15 ± 3.02** | - |
| Qwen3-VL-8B | edv_esv | 8.02 ± 2.49 | 7.06 ± 2.83 | 28.15 ± 18.24 |
| | esv_edv | **7.12 ± 3.90** | **10.32 ± 4.40** | **36.57 ± 85.62** |
| | edv | **6.55 ± 2.59** | - | - |
| | esv | - | 11.34 ± 5.82 | - |
| Qwen3-VL-32B | edv_esv | 10.05 ± 3.09 | 7.83 ± 2.06 | **25.73 ± 13.82** |
| | esv_edv | 12.13 ± 2.88 | 18.21 ± 7.38 | 60.20 ± 105.97 |
| | edv | 6.64 ± 2.08 | - | - |
| | esv | - | 11.22 ± 4.14 | - |
| EchoCLIP | - | - | - | 8.54 ± 10.07 |

Table 5: Metrics of model performance in T1 original showing evaluation uncertainty (sampling variance). The standard deviations of all MAEs shown here were calculated across videos. We highlighted the best model performance in different context conditions.

| Model | Output format | EDV (frame) MAE↓ | ESV (frame) MAE↓ | EF (%) EF MAE↓ |
|---|---|---|---|---|
| Random | - | 11.87 | 11.62 | - |
| LLaVA-Interleave | edv_esv | 14.11 ± 11.02 | 11.71 ± 3.05 | 49.40 ± 20.19 |
| | esv_edv | 23.64 ± 27.13 | 18.12 ± 22.94 | 62.23 ± 24.24 |
| | edv | 18.28 ± 41.38 | - | - |
| | esv | - | 22.00 ± 79.02 | - |
| LLaVA-NeXT-Video-7B | edv_esv | 34.25 ± 23.14 | 20.92 ± 11.52 | 45.62 ± 28.08 |
| | esv_edv | 28.22 ± 20.69 | 24.08 ± 17.56 | 46.33 ± 16.93 |
| | edv | 27.52 ± 16.75 | - | - |
| | esv | - | 26.04 ± 16.77 | - |
| LLaVA-NeXT-Video-34B | edv_esv | 9.95 ± 2.57 | **6.65 ± 2.00** | 38.85 ± 21.21 |
| | esv_edv | 18.50 ± 5.01 | 11.96 ± 4.57 | 67.62 ± 31.04 |
| | edv | 10.63 ± 2.85 | - | - |
| | esv | - | 11.04 ± 4.51 | - |
| Gemma-3n | edv_esv | 10.04 ± 3.24 | 14.69 ± 5.94 | 58.56 ± 25.48 |
| | esv_edv | 12.62 ± 8.14 | **7.05 ± 6.91** | **35.39 ± 22.42** |
| | edv | 13.50 ± 3.41 | - | - |
| | esv | - | **6.20 ± 2.97** | - |
| Qwen3-VL-8B | edv_esv | **9.27 ± 2.71** | 11.10 ± 2.90 | **31.63 ± 17.68** |
| | esv_edv | 15.57 ± 5.50 | 20.19 ± 6.53 | 48.88 ± 22.60 |
| | edv | **9.48 ± 2.85** | - | - |
| | esv | - | 20.02 ± 8.67 | - |
| Qwen3-VL-32B | edv_esv | 10.54 ± 3.01 | 16.36 ± 8.68 | 31.81 ± 14.63 |
| | esv_edv | **10.11 ± 3.57** | 27.19 ± 6.12 | 55.90 ± 83.04 |
| | edv | 10.48 ± 2.96 | - | - |
| | esv | - | 21.78 ± 8.89 | - |

Table 6: Metrics of model performance in T1 segmented showing evaluation uncertainty (sampling variance). The standard deviations of all MAEs shown here were calculated across videos. The best model performance among each output format was highlighted.

| | | EDV (frame) | ESV (frame) | EF (%) |
|---|---|---|---|---|
| **Model** | **Output format** | **MAE↓** | **MAE↓** | **MAE↓** |
| Random | - | 11.87 | 11.62 | - |
| LLaVA-Interleave | large_small | 12.08 ± 3.69 | 14.73 ± 4.02 | 53.26 ± 20.87 |
| | small_large | 20.45 ± 7.20 | 14.45 ± 4.48 | 59.70 ± 23.43 |
| | large | 12.24 ± 5.39 | - | - |
| | small | - | 10.03 ± 2.87 | - |
| LLaVA-NeXT-Video-7B | large_small | 30.71 ± 20.05 | 14.24 ± 5.90 | 54.13 ± 25.94 |
| | small_large | **15.52 ± 6.30** | 19.44 ± 7.59 | 61.22 ± 57.09 |
| | large | 36.95 ± 23.73 | - | - |
| | small | - | 19.25 ± 12.57 | - |
| LLaVA-NeXT-Video-34B | large_small | 11.02 ± 2.98 | **10.83 ± 3.70** | 52.56 ± 51.65 |
| | small_large | 17.82 ± 4.85 | 13.33 ± 5.08 | 66.70 ± 41.54 |
| | large | 12.03 ± 2.96 | - | - |
| | small | - | 12.00 ± 4.92 | - |
| Gemma-3n | large_small | 15.15 ± 3.74 | 15.87 ± 4.47 | 91.34 ± 36.29 |
| | small_large | 23.42 ± 5.79 | **10.71 ± 4.99** | 63.70 ± 45.20 |
| | large | 18.69 ± 4.05 | - | - |
| | small | - | **2.95 ± 1.98** | - |
| Qwen3-VL-8B | large_small | 10.54 ± 3.01 | 11.28 ± 3.52 | **31.39 ± 17.79** |
| | small_large | 27.71 ± 6.53 | 27.40 ± 6.26 | 55.71 ± 83.37 |
| | large | 10.54 ± 3.01 | - | - |
| | small | - | 26.08 ± 6.34 | - |
| Qwen3-VL-32B | large_small | **10.52 ± 3.01** | 23.79 ± 7.59 | 37.00 ± 12.60 |
| | small_large | 18.20 ± 8.10 | 27.47 ± 6.16 | **54.36 ± 134.16** |
| | large | **10.44 ± 2.95** | - | - |
| | small | - | 27.39 ± 6.11 | - |

Table 7: Metrics of model performance in T1 non-medical showing evaluation uncertainty (sampling variance). The standard deviations of all MAEs shown here were calculated across videos. The best model performance among each output format was highlighted.

## Appendix B. Plots of predicted vs ground truth index from Gemma-3n outputs

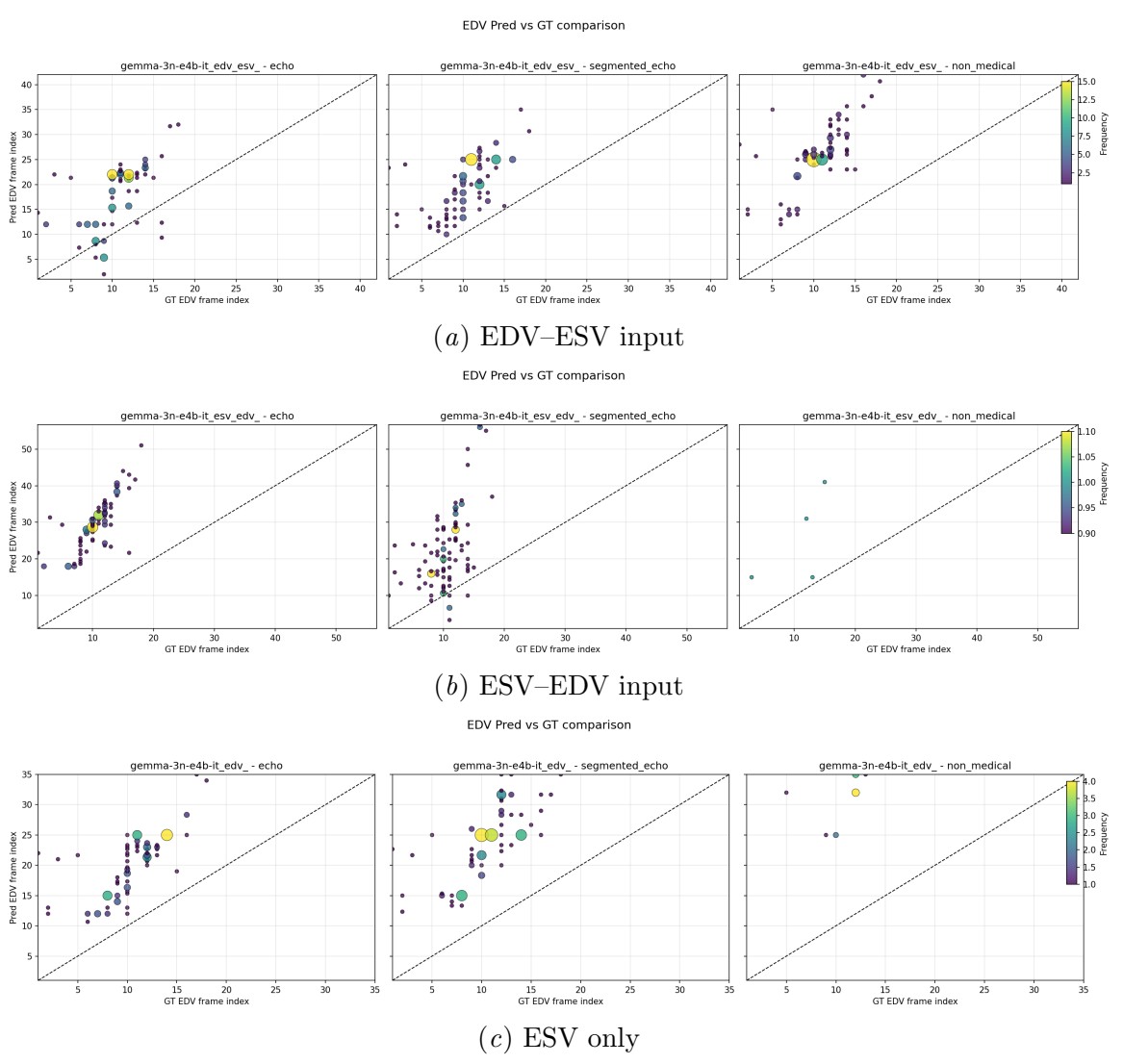

(*a*) EDV–ESV input

(*b*) ESV–EDV input

(*c*) ESV only

Figure 6: Additional EDV examples for Gemma-3n

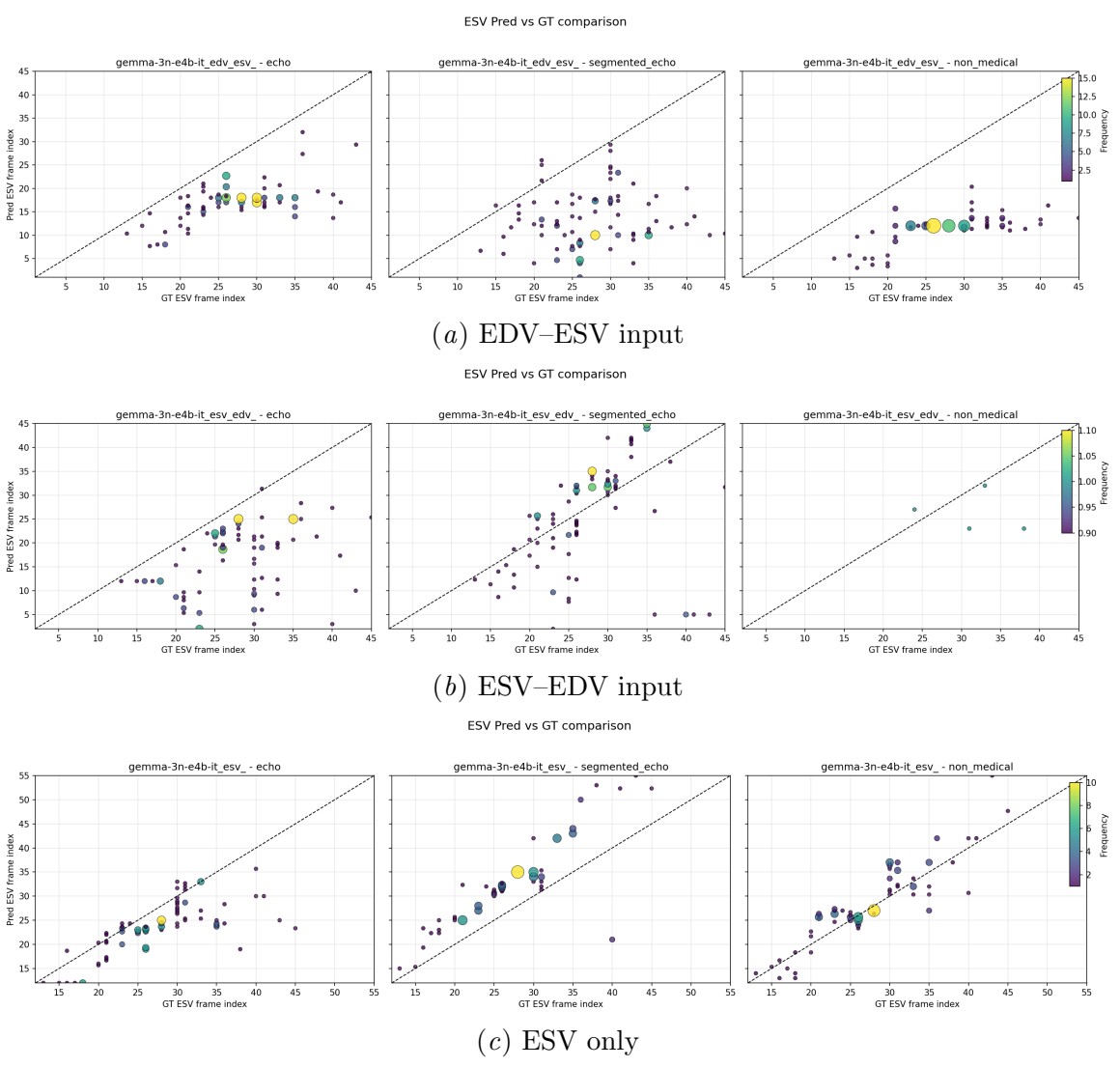

(a) EDV–ESV input

(b) ESV–EDV input

(c) ESV only

Figure 7: Additional ESV examples for Gemma-3n

