# OpenReview forum: "Can Vision Language Models Track a Heartbeat? A Benchmark on Frame-Level Echocardiogram Understanding"
_MIDL.io/2026/Validation_Papers — MIDL 2026 - Validation Papers Poster_

### Official Review · Reviewer_6Sya · 2026-01-03

**Confidence:** 4
**Preliminary Rating:** 4
**Final Rating:** 5

**Summary:**

This research addresses a critical clinical question: whether state-of-the-art open-source Vision Language Models (VLMs) can perform frame-level temporal reasoning for echocardiogram interpretation—essential for calculating ejection fraction (EF) via accurate localization of end-diastole (EDV) and end-systole (ESV) frames. Using 100 videos from the EchoNet-Dynamic dataset, the authors designed two progressive tasks: a primary frame localization task (T1) to identify EDV/ESV indices from cardiac cycle subsequences and an auxiliary two-frame classification task (T2) simplifying to size comparison, with three input conditions (original videos, left ventricular segmentation overlays, non-medical segmented videos) and Monte Carlo random baselines. Six leading open-source VLMs (e.g., Gemma 3n, LLaVA-NeXT-Video 7B/34B, Qwen3-VL 8B/32B) were evaluated, showing all models performed poorly on T1 with errors far exceeding clinical tolerances (2-3 frames) and often near random levels, while only Qwen3-VL series achieved >0.9 accuracy on T2 with segmentation and non-medical cues. As the first systematic evaluation of general-purpose VLMs on frame-level echocardiogram analysis, this work reveals limitations in their handling of fine-grained temporal and spatial dynamics in medical videos, highlights the need for medical-specific benchmarks and domain-adapted temporal modeling for future medical VLMs.

**Strengths:**

1.The research targets a clinically critical task (EDV/ESV frame localization for EF calculation), addressing the time-consuming manual work in clinical practice and boasting strong translational relevance.
2. The research progressive experimental design (T1 frame localization + T2 two-frame classification) with three input conditions and Monte Carlo random baselines enables rigorous, layered analysis of model limitations.

**Weaknesses:**

The experiments are solely based on a subset of 100 videos from the EchoNet-Dynamic dataset (randomly selected from the dataset's validation set) without any external validation.
The study only focuses on technical performance metrics (e.g., MAE, accuracy) of models in EDV/ESV frame localization and two-frame discrimination tasks, lacking clinical utility assessment.

**Detailed Comments:**

In Sec. 2.2, the authors fail to explicitly define the criteria for determining a “cardiac cycle” or the subsequence sampling method.
The description of the three prompt variants in Sec. 2.2 is ambiguous, as it does not clarify whether the semantic content (e.g., EDV/ESV explanations) is consistent across variants.
Tables 1-3 lack clear annotations for the “Output format” column and do not specify whether MAE values are averaged across seeds/passes or include variability measures.
The authors attribute Gemma-3n’s low ESV-only MAE to “mid-range bias” in Sec. 3.1 but provide insufficient quantitative/visual evidence and fail to explain why this bias is specific to ESV-only prediction.
Sec. 2.4 incompletely describes model input formats, as it does not specify the number of sampled frames for multi-image inputs.

**Justification Of Final Rating:**

Most of my previous concerns have been addressed, and the author has provided satisfactory responses and revisions, which have enriched the content of the paper as a whole. I therefore adjust my rating to 5.

**Justification Of The Preliminary Rating:**

This research covers an interesting topic. However, its methodology and experiments have critical ambiguities, e.g., undefined cardiac cycle criteria and vague prompt descriptions. The "mid-range bias" explanation for Gemma-3n’s poor performance lacks evidence. Key details like multi-image frame counts are also unspecified. These flaws need revisions, but the paper’s potential remains. Thus, I give a weak accept.

**Questions To Address In The Rebuttal:**

Please check the Weaknesses and Detailed Comments.

---

### Official Review · Reviewer_M7Xh · 2026-01-09

**Confidence:** 4
**Preliminary Rating:** 3

**Summary:**

This paper proposes and systematically constructs a Heartbeat Tracking Benchmark, targeting the fundamental yet long-standing underexplored task of heartbeat / cardiac cycle tracking in cardiac imaging, for which no unified evaluation standard has previously existed. The benchmark integrates datasets, task definitions, evaluation metrics, and comparative experiments across multiple representative methods. The proposed benchmark is of notable clinical relevance, and the topic aligns well with the scope of the Validation Track.

**Strengths:**

1. Heartbeat cycle tracking is a key intermediate task in cardiac imaging analysis and plays an important role in downstream clinical analysis and diagnosis.

2. The data splits, evaluation protocols, and experimental settings are clearly described and well justified.

3. The benchmark is composed of multiple tasks with varying levels of difficulty, which enables a more fine-grained and comprehensive comparison of model performance.

**Weaknesses:**

1. The LLMs evaluated in this work are all general-purpose models that are not specifically designed for medical applications. The experiments would be more convincing if medical-domain LLMs and echocardiography-specific LLMs (e.g., EchoPrime) were included.

2. The discussion section could be strengthened by providing deeper insights into the implications for future LLM development, rather than merely summarizing the current experimental results.

**Detailed Comments:**

Refer to weaknesses.

**Justification Of The Preliminary Rating:**

As a benchmark paper, this work is solid in terms of task definition, experimental design, and result presentation, and it provides clear value to the community. However, the selection of evaluated models is not sufficiently rigorous. It is well known that general open-source LLMs have limited performance across many medical tasks, so their suboptimal performance in this setting is not surprising. In addition, the paper lacks a forward-looking analysis and concrete recommendations for future medical LLM development.

**Questions To Address In The Rebuttal:**

Refer to weaknesses.

---

### Official Review · Reviewer_cX9o · 2026-01-09

**Confidence:** 3
**Preliminary Rating:** 3
**Final Rating:** 4

**Summary:**

The authors investigate the use of general-purpose vision language models (VLMs) for a temporal medical imaging problem. Using cardiac echocardiography videos, they evaluate six contemporary VLMs on frame localization of end-diastole and end-systole (EDV/ESV) via task-specific prompting under multiple contexts, without any fine-tuning/training of VLMs on echo data. Across these settings, the results suggest that current general-purpose VLMs struggle to reliably reason about periodic cardiac motion and temporal function in echocardiography. Overall, the study provides a validation of current VLM capabilities in this domain and motivates the need for more specialized, medically curated VLMs for robust cardiovascular applications.

**Strengths:**

- **Clinical Relevance**: The paper tackles a clinically relevant question: how well current general-purpose VLMs transfer to temporal echocardiography, without any fine-tuning of VLMs on echo data. The findings highlight important limitations of current VLMs in reasoning about periodic physiological motion when applied directly to medical imaging domain, motivating domain-specific temporal modeling for future medical VLMs.
- **Reproducibility support**: The authors provide strong reproducibility support, e.g., code, prompts, and data, are all available online.

**Weaknesses:**

- **Limited Study Scope**: The evaluation is thorough (six VLMs, two tasks, multiple prompting/context settings on 100 echocardiography videos), but the consistently poor performance and observed failure modes (e.g., strong order bias, out-of-range index errors, low-utility segmentation masks) demand deeper analysis. The study could be expanded by performing additional experiments e.g., by including lightweight re-training/fine-tuning of VLM on a bigger public echo dataset to improve performance, or by improving temporal reasoning by modeling periodic behavior for echo data in VLM. These developments would help to clarify whether the observed limitations are mainly due to domain shift, missing temporal modeling, or both.
- **Practical Limitations**: The current setup has several practical limitations, including prompt sensitivity/need for optimization, evaluation on a single dataset with a single view/orientation, and a model architecture not tailored to periodic cardiac motion.
- **Foundation echo VLMs**: Since echocardiography-trained foundation models have been reported in the literature (e.g., [1], [2]) with strong performance, it is unclear why the study focuses only on general-purpose VLMs.
- **Missing qualitative visualization**: The manuscript would benefit from at least one representative echocardiography figure to help readers unfamiliar with echocardiography understand the data and the visual cues relevant to EDV/ESV localization.

[1] Kim, S., Jin, P., Song, S., Chen, C., Li, Y., Ren, H., ... & Li, Q. (2025). Echofm: Foundation model for generalizable echocardiogram analysis. IEEE transactions on medical imaging.

[2] Christensen, M., Vukadinovic, M., Yuan, N., & Ouyang, D. (2024). Vision–language foundation model for echocardiogram interpretation. Nature Medicine, 30(5), 1481-1488.

**Detailed Comments:**

- Please report variability across videos by adding standard deviation alongside mean in Tables 1–3.
- Draw a small schematic figure to illustrate pipeline on high-level: context/prompt → VLM model → frame task → Volume estimation/metrics.

**Justification Of Final Rating:**

The authors’ rebuttal and revised manuscript address my main concerns. They added a clearer motivation for focusing on general-purpose VLMs, improved qualitative understanding by including representative echocardiography examples and a pipeline schematic, and strengthened the experimental reporting by adding variability analyses across videos. While the paper still does not include adaptation or temporal modeling experiments, the revised manuscript more clearly positions this as a limitation and a future direction. Overall, the work provides a useful and reproducible cautionary evaluation of current general-purpose VLM behavior on temporal echocardiography tasks, and I update my rating to 4.

**Justification Of The Preliminary Rating:**

The paper has some validation-related strengths such as clear clinical relevance and reproducibility support. However, it lacks necessary experiments, sufficient literature review, and provides incomplete results.

**Questions To Address In The Rebuttal:**

Address weaknesses
- Perform additional experiments by including lightweight re-training/fine-tuning of VLM on a bigger public echo dataset to improve performance, or by improving temporal reasoning by modeling periodic cardiac behavior for temporal echo data in VLM.
- Cite echo VLMs and explain the motivation to use general-purpose VLMs when echo-trained VLMs are available.
- The manuscript would benefit from at least one representative echocardiography figure to help readers unfamiliar with echocardiography understand the data and the visual cues relevant to EDV/ESV localization.
- Please report variability across videos by adding standard deviation alongside mean in Tables 1–3.
- Draw a small schematic figure to illustrate pipeline on high-level: context/prompt → VLM model → frame task → Volume estimation/metrics.

---

### Author Rebuttal · Authors · 2026-01-24

**Rebuttal:**

We appreciate the reviewers’ thoughtful feedback, which has helped us to strengthen our manuscript. Below we summarize the revisions made in response, and we have uploaded a revised manuscript with all changes highlighted:

1. **Added Related Work (Section 2).** We added a dedicated Related Work section, discussing echocardiography-trained models and clarifying why our benchmark mainly focuses on general-purpose VLMs.

2. **Added Task and Data Overview Figure.** We introduced an overview plot that includes a high-level pipeline flowchart, representative EDV and ESV frames, and examples of segmentation masks produced by our segmentation step.

3. **Clarified Cardiac Cycle and Subsequence Extraction (Section 3.2).** We refined the definition of a cardiac cycle and described in more detail how we extract the subsequence used for Task 1. We also added statistics describing subsequence lengths.

4. **Emphasized Prompt Consistency Across Contexts (Section 3.2).** To make our “maximally consistent prompting” strategy explicit, we added a clearer description of the prompts used across the three context conditions.

5. **Added EchoCLIP EF Baseline (Section 3.4, Table 1).** We ran EchoCLIP on the original videos for ejection fraction prediction and report its EF MAE in Table 1 as a non-conversational reference and an approximate upper bound on performance for task T1 original.

6. **Updated Tables 1–3 With Variability Measures (Table 1-3, Appendix A).** We revised Tables 1–3 to report mean ± standard deviation (across random seeds) to reflect model stochasticity. Additional tables showing across-video variability are placed in Appendix A.

7. **Clarified “Output format” (Section 4.1).** We added an explicit explanation of the “Output format” column used in Tables 1–3.

8. **Added Qualitative Evidence for Gemma-3n Behavior (Section 4.1, Appendix B).** We included a new example figure illustrating Gemma-3n’s tendency to respond with mid-range indices. Full qualitative results are provided in Appendix B.

9. **Strengthened the Discussion.** We revised the Discussion section to better articulate limitations and to outline concrete future directions for improving the reliability of general-purpose VLMs on echocardiography video understanding tasks.

**Supporting Material:**

/attachment/39c1a5428b70d218004b9bff0fac8955e2d9443d.pdf

---

### Meta-Review · Area_Chair_wo4i · 2026-02-09

**Recommendation:** Accept (Poster)
**Confidence:** 4

**Metareview:**

The paper presents a clinically motivated benchmark for frame-level EDV/ESV localization in apical four-chamber echocardiograms and evaluates six general-purpose VLMs under consistent prompting and multiple input conditions, with Monte Carlo baselines. Two reviewers are positive after the rebuttal, while one reviewer remains borderline and raises a valid concern that the results on general-purpose VLMs are not surprising without stronger medical or echocardiography-specific conversational baselines or adaptation experiments. The rebuttal addresses much of the methodological ambiguity and better positions the study as an out-of-the-box evaluation, with an additional EchoCLIP EF baseline added during the rebuttal. Overall, I believe the benchmark is valuable to the validation track.

---

### Decision · Program_Chairs · 2026-02-14

Accept (Poster)